# Axon terminals control endolysosome diffusion to support synaptic remodelling

Beatrice Terni[1,2] , Artur Llobet[1,2]

Endolysosomes are acidic organelles formed by the fusion of endosomes with lysosomes. In the presynaptic compartment they contribute to protein homeostasis, the maintenance of vesicle pools and synaptic stability. Here, we evaluated the mobility of endolysosomes found in axon terminals of olfactory sensory neurons of *Xenopus tropicalis* tadpoles. F-actin restricts the motion of these presynaptic acidic organelles which is characterized by a diffusion coefficient of $6.7 \times 10^{-3}\ \mu m^2 \cdot s^{-1}$. Local injection of secreted protein acidic and rich in cysteine (SPARC) in the glomerular layer of the olfactory bulb disrupts the structure of synaptic F-actin patches and increases the presence and mobility of endolysosomal organelles found in axon terminals. The increased motion of endolysosomes is localized to the presynaptic compartment and does not promote their access to axonal regions for retrograde transportation to the cell body. Local activation of synaptic degradation mechanisms mediated by SPARC coincides with a loss of the ability of tadpoles to detect waterborne odorants. Together, these observations show that the diffusion of presynaptic endolysosomes increases during conditions of synaptic remodelling to support their local degradative activity.

## Introduction

The transfer of information in the nervous system occurs when presynaptic axon terminals release neurotransmitters to bind specific postsynaptic receptors. This highly regulated process relies on the tight control of membrane trafficking by the presynaptic compartment. Axon terminals finely adjust the secretion of neurotransmitters through the modification of the number, location, and molecular composition of synaptic vesicles. Although most of our knowledge on the adaptive changes in neurotransmitter release comes from studies on synaptic vesicles, there is growing evidence pointing to the importance of other organelles. Endolysosomes are an example (Huotari & Helenius, 2011). They form as a result of the fusion of lysosomes

with endosomes and contribute to protein homeostasis (Azarnia Tehran et al, 2018), the formation and maintenance of vesicle pools (Di Giovanni & Sheng, 2015), and synaptic stability (Winckler et al, 2018). Despite these observations, many of the relationships established among the organelles of the endolysosomal system (Klumperman & Raposo, 2014) within the presynaptic compartment remain to be characterized.

The constant interplay among cytoplasmic organelles is determined by their mobility. For example, confined diffusion of synaptic vesicles in hippocampal boutons is characterized by coefficients that typically range from $5 \times 10^{-5}\ \mu m^2 \cdot s^{-1}$ (Jordan et al, 2005) to ~$1 \times 10^{-4}\ \mu m^2 \cdot s^{-1}$ (Forte et al, 2017). But diffusion coefficients increase to ~$0.01\ \mu m^2 \cdot s^{-1}$ in synapses that must sustain higher neurotransmitter release rates such as retinal bipolar cell terminals or cerebellar mossy fibre terminals (Holt et al, 2004; Rothman et al, 2016). A yet open question is if mobility of the organelles that constitute the endolysosomal system is also adapted to meet synaptic demands. The function of endolysosomes is balanced between a homeostatic role, aiming to maintain the correct protein composition of the presynaptic terminal, and a net degradative role, committed to the elimination of macromolecular complexes. The homeostatic role of endolysosomes can be associated to autophagy and is considered part of the constitutive processes taking place in the presynaptic compartment (Vijayan & Verstreken, 2017). In contrast, a predominant degradative role only becomes evident during certain circumstances, for instance, when there is massive elimination of synaptic contacts in postnatal development (Song et al, 2008). It is thus necessary to find out in a biologically relevant context whether the shift in the equilibrium between homeostatic and degradative roles of endolysosomal organelles is associated to a change in their mobility.

*Xenopus tropicalis* tadpoles offer a unique experimental platform to address this question because they allow the in vivo visualization of the entire morphology of some neuronal types, as for example, olfactory sensory neurons (OSNs). These bipolar neurons have their soma located in the olfactory epithelium, extend their axons through the olfactory nerve and their terminals configure the presynaptic element of the glomeruli found in the olfactory bulb (Nezlin et al, 2003; Terni et al, 2017). Moreover, the metabolic activity

[1]Department of Pathology and Experimental Therapy, School of Medicine, Institute of Neurosciences, University of Barcelona, Barcelona, Spain   [2]Laboratory of Neurobiology, Bellvitge Biomedical Research Institute (IDIBELL), Barcelona, Spain

Correspondence: beatriceterni@ub.edu; allobet@ub.edu

of OSNs is optimal at room temperature because *X. tropicalis* tadpoles develop between 24°C and 30°C. All these advantages allowed us to track the mobility of endolysosomes stained with lysotracker deep red in a subpopulation of genetically labelled OSNs. Endolysosomes move exclusively by confined diffusion in the presynaptic compartment to support a homeostatic role. During the cell-autonomous remodelling of synaptic terminals induced by secreted protein acidic and rich in cysteine (SPARC; López-Murcia et al, 2015), their diffusion coefficient increases by a factor of two. This change is restricted to the presynaptic compartment, thus suggesting axon terminals locally adjust the diffusion of endolysosomes to meet their activity demands.

## Results

### In vivo visualization of endolysosomes in OSNs of *X. tropicalis* tadpoles

OSN axons leave the olfactory nerve and enter into the olfactory bulb to form the nerve layer and the glomerular layer. Most of the organelles of the endolysosomal system contain LAMP-1, a transmembrane protein found in lysosomes (Luzio et al, 2014) and in neuronal endocytic organelles (Cheng et al, 2018). Expression of LAMP-1 was found in OSN axons, as well as in presynaptic terminals, which were identified by the synaptic vesicle marker synaptophysin (Fig 1A). The characteristic morphology of olfactory glomeruli was evident both by immunohistochemistry and transmission electron microscopy (Fig 1B). OSN axon terminals defined a large, tortuous, presynaptic compartment that contained a high density of synaptic vesicles and multiple mitochondria. Active zones indicated the establishment of synaptic contacts on dendrites of mitral/tufted cells (Fig 1C). Elements of the endolysosomal system, such as late endosomes (Fig 1C) or lysosomes (Fig 1D) were obvious in the presynaptic compartment. Occasionally, some autophagic shapes were also observed (Fig 1E), indicating that autophagy and the endolysosomal system play a role in the normal function of presynaptic terminals of OSNs.

The olfactory system is wired according to the expression of odorant receptors in OSNs (Buck & Axel, 1991) so that all neurons containing the same receptor send their axons to a defined glomerulus (Mombaerts, 2006). The use of the *zHB9::GFP X. tropicalis* transgenic line allowed visualization of the entire morphology of a subset of OSNs (Fig 1F). Fluorescently labelled cell bodies were sparsely distributed throughout the olfactory epithelium. OSNs sent their axons along the olfactory nerves, entered in the olfactory bulb and gave rise to a glomerular structure. Projection of all GFP-positive axons into a defined glomerulus indicated that labelled OSNs were a homogeneous neuronal population expressing a common, yet unknown, olfactory receptor.

Local application of lysotracker deep red in the nasal cavity of *zHB9::GFP X. tropicalis* tadpoles allowed the visualization of acidic organelles in all compartments of OSNs as a result of transportation from the cell body to axon terminals (see the Materials and Methods section for details). Endolysosomes are the principal organelles with

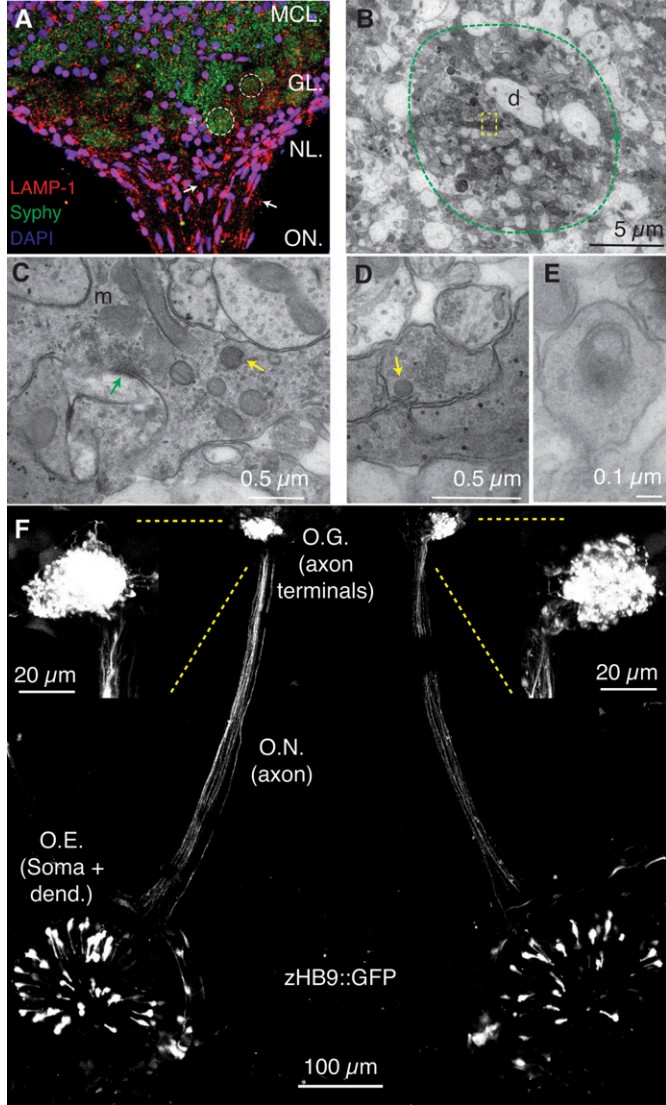

**Figure 1. The endolysosomal system is present in axon terminals of olfactory sensory neurons (OSNs) of *Xenopus tropicalis* tadpoles.**
**(A)** Image of the olfactory bulb of a *X. tropicalis* tadpole stained with LAMP-1, an endolysosomal marker. LAMP-1 was present in axons (arrows) of OSNs entering the olfactory bulb, as well as in olfactory glomeruli, which were revealed by the accumulation of the synaptic vesicle protein synaptophysin (dotted circles). Axon terminals of OSNs form the presynaptic element of olfactory glomeruli. DAPI was used to reveal the position of cell nuclei. The image shows three regions of the olfactory bulb: nerve layer, glomerular layer, and the rostral portion of the mitral cell layer. Part of the olfactory nerve is also visible. **(B)**. Transmission electron microscopy images allowed the observation of olfactory glomeruli (dotted circle). Axon terminals appeared as electrodense processes contacting dendrites (d). **(C)** The presynaptic compartment of OSNs is long, tortuous, and contains a large pool of cytoplasmic vesicles that acts as a reservoir of active zones (green arrow). Mitochondria (m) and late endosomes (yellow arrow) are present in the cytosol. **(D)** Detail of the boxed region in (B) indicating the presence of a lysosome (arrow). **(E)** Some autophagophores were also present in OSN presynaptic terminals. **(F)** Visualization of the entire morphology of OSNs in a *zHB9::GFP X. tropicalis* tadpole anesthetized and embedded in agarose. It is possible to observe the entire morphology of a population of GFP positive OSNs that project to a single glomerulus. Cell bodies are located in the olfactory epithelium (O.E). Axons travel along the olfactory nerve and give rise to two symmetrical olfactory glomeruli (O.G.) in the olfactory bulbs.

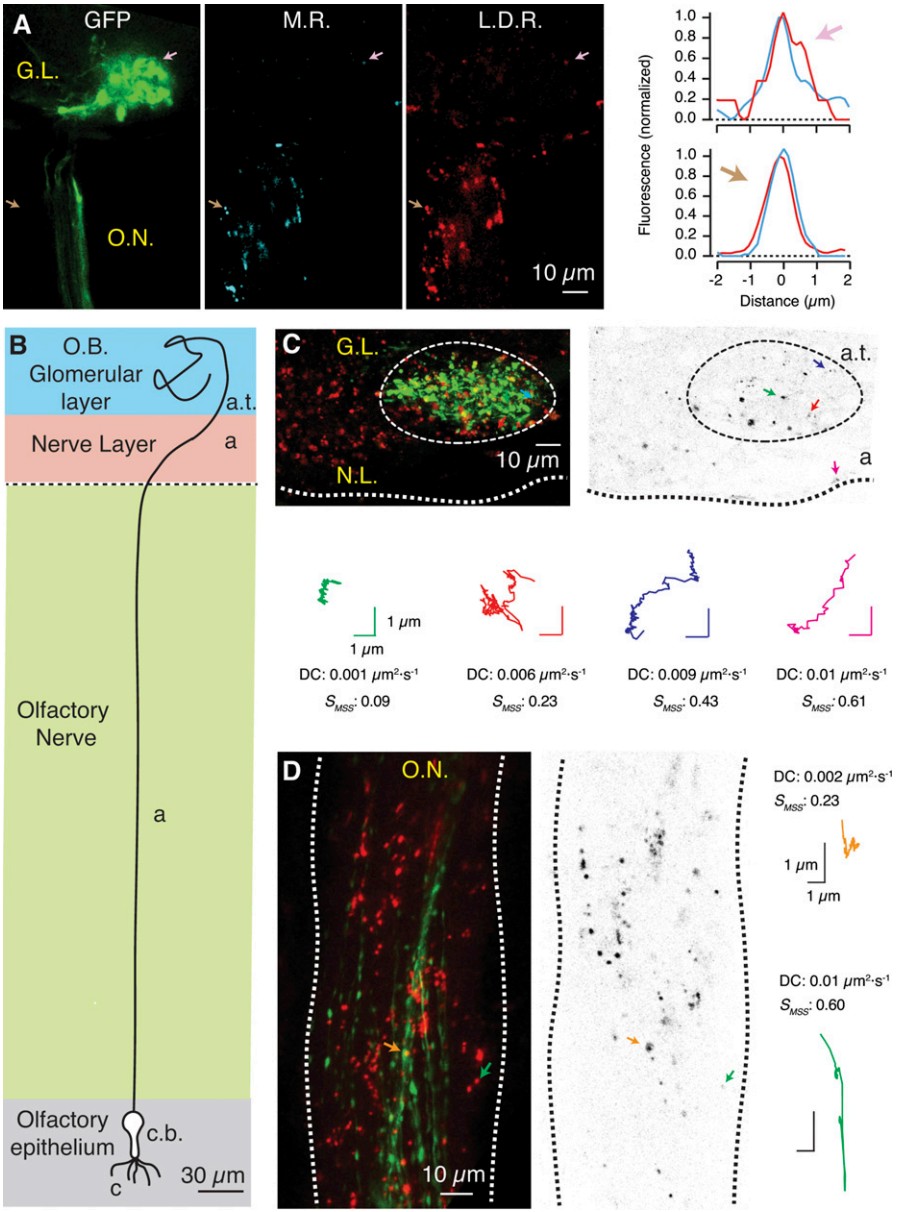

**Figure 2. Tracking the mobility of endolysosomes in olfactory sensory neurons in vivo.**
**(A)** Puncta stained with lysotracker deep red (L.D.R, red) co-localized with the product of the hydrolysis of magic red cathepsin-B (M.R., blue). The arrows indicate the line profiles of two acidic organelles located in the periphery of the GFP labelled glomerulus (arrow, magenta) and in an axonal region (arrow, brown). **(B)** Schematic morphology of a *Xenopus tropicalis* tadpole olfactory sensory neuron. The anatomical regions are indicated. Cilia (c), cell body (c.b.), axonal region (a) which comprises the olfactory nerve (ON) tract and the nerve layer (NL) of the olfactory bulb, and axon terminals forming glomeruli (a.t.). **(C)** Maximum intensity projection (left) and single confocal section (right) of the olfactory bulb. The mobility of acidic organelles (red staining) was evaluated separately in axon terminals originating the GFP-positive glomerulus (dotted circle) and in the NL of the olfactory bulb. The $S_{MSS}$ and the diffusion coefficient were calculated for puncta that fulfilled the analysis criteria (see the Materials and Methods section for details). Mobility tracks are indicated for three glomerular acidic organelles (red, green, and blue traces) and one acidic organelle of the NL (magenta). **(D)** Maximum intensity projection (left) and single confocal section (right) of a region of an ON. Acidic organelles found in the ON moved by confined diffusion (orange trace) and directed motion (green trace). Notice the presence of a population of GFP labelled axons, a characteristic of the *zHB9::GFP X. tropicalis* line.

hydrolase activity and consequently, they accumulate acidotropic probes such as lysotracker dyes. Particles labelled with lysotracker deep red were also stained for the product of magic red cathepsin-B (Fig 2A and Video 1) that accumulates in organelles containing catalytically active cathepsin-B (Creasy et al, 2007). The association of an active proteolytic activity to an acidic pH confirmed the identity of particles labelled with lysotracker deep red as endolysosomes (Bright et al, 2016).

## Quantification of endolysosomal mobility in neuronal compartments

Endolysosomal movement was visualized in different neuronal compartments (Figs 1F and 2A and B) of tadpoles anesthetized and embedded in agarose. Experimental viability was certified by

observing the blood flow in vessels running along the olfactory nerve (Video 2). This control guaranteed that the nervous system received a correct energy supply during experimental time, ruling out the contribution of possible changes in the mobility of endolysosomal organelles caused by metabolic alterations.

The moment scaling spectrum (MSS) was used to quantitatively characterize particle mobility (Sbalzarini & Koumoutsakos, 2005). The slope of the MSS, defined as $S_{MSS}$, can appear in values ranging from 0 to 1. $S_{MSS} = 0.5$ represents free diffusion, $0 < S_{MSS} < 0.5$ is associated to confined diffusion and $0.5 < S_{MSS} < 1$ indicates directed motion. Presynaptic endolysosomes found in GFP labelled glomerulus showed $S_{MSS}$ values representative of confined diffusion (Fig 2C and Video 3) with a diffusivity ranging between 0.001 and 0.01 $\mu m^2 \cdot s^{-1}$. To investigate if the observed mobility was specific of presynaptic terminals, we analysed axonal

regions found both at the level of the olfactory bulb and in the olfactory nerve. Movement of endolysosomal organelles in the nerve layer of the olfactory bulb was still described by confined diffusion but some $S_{MSS}$ values were around 0.5 (Fig 2C, magenta trace). In the olfactory nerve, the mobility was qualitatively similar to the nerve layer of the olfactory bulb. Most of the trajectories exhibited restricted diffusion; however, fast movements characterized by $S_{MSS}$ > 0.5 were obvious (Video 4). For example, the particle shown in Fig 2D (green trace) moved a distance of 10 $\mu$m reaching a maximum speed of 1.2 $\mu$m·s$^{-1}$, which are values observed for the motion of endosomes in association to microtubules (Granger et al, 2014).

$S_{MSS}$ values were plotted as a function of the associated diffusion coefficients in the three neuronal regions investigated (Fig 3A–D). This is a useful approach to characterize the movement of single particles in cells (Ewers et al, 2005; Arts et al, 2019). Endolysosomes found in GFP-positive presynaptic terminals forming a unique glomerulus moved by confined diffusion with an average diffusivity of 6.7 × 10$^{-3}$ $\mu$m$^2$·s$^{-1}$ (n = 82, seven tadpoles, Fig 3A and D). Directed movements were not detected, a result that could be attributed to the limitation to observe transient interactions between microtubules and endolysosomes. This possibility was explored by investigating mobility in axonal regions, which are a bona fide control

to observe active transport of organelles. The majority of the particles found in the nerve layer of the olfactory bulb still moved by confined diffusion with an average diffusion coefficient of 7.4 × 10$^{-3}$ $\mu$m$^2$·s$^{-1}$ (n = 40, seven tadpoles); however, 15% of trajectories approximated to free diffusion or even to directed movement (Fig 3B and D). The short length of axons forming the nerve layer restricted the number of observations, a limitation that was not present in the central region of the olfactory nerve (Fig 3C). Here, a small proportion of axonal particles showed directed motion, which correponded to endolysosomal organelles that reached a maximum velocity of 1 ± 0.8 $\mu$m·s$^{-1}$ in trajectories measuring 10 ± 1.5 $\mu$m (n = 4) and likely reflected mobility along microtubules. Moreover, $S_{MSS}$ values found between 0.4 and 0.6, consistent with the presence of random movements (free diffusion), were observed in 11 trajectories (Fig 3C). The majority of the trajectories (88%) were yet associated to confined diffusion (n = 118, seven tadpoles), which contributed to define a diffusion coefficient of 8.3 × 10$^{-3}$ $\mu$m$^2$·s$^{-1}$ (Fig 3D).

The 3% of endolysosomes detected at the axonal level (nerve layer of the olfactory bulb and olfactory nerve) that showed directed movement was likely an underestimation. Because the accuracy of $S_{MSS}$ values is compromised in short tracks (Vega et al, 2018), we only analysed the mobility of particles that could be

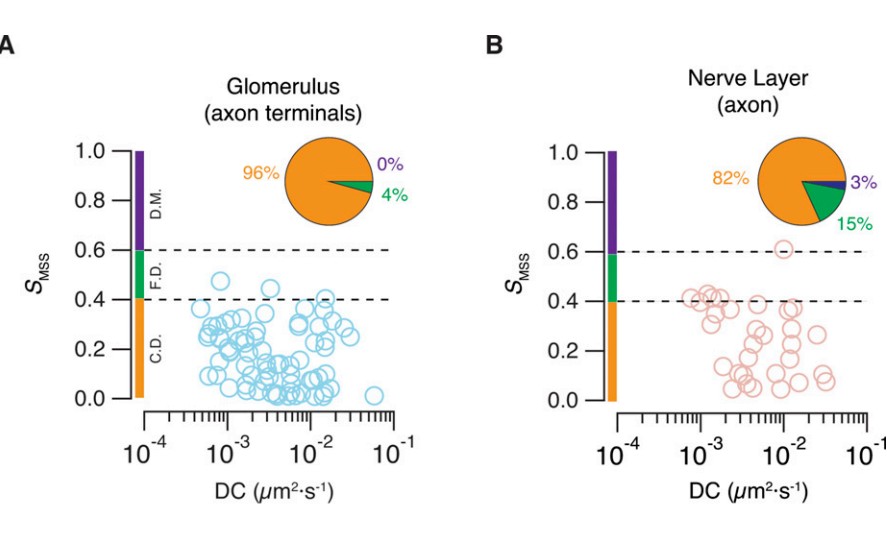

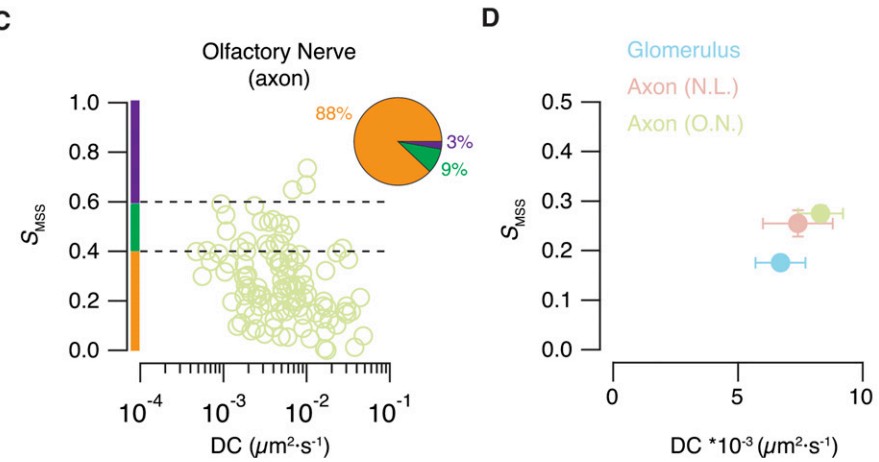

**Figure 3. Mobility of endolysosomes found in presynaptic and axonal compartments.**
**(A, B, C)** Plot of the slope of the moment scaling spectrum ($S_{MSS}$) against the diffusion coefficient (D.C.) of acidic organelles located in three different layers of the olfactory system. $S_{MSS}$ values found between 0.4 and 0.6 are associated to free diffusion (F.D., dotted lines), whereas $S_{MSS}$ values <0.4 or >0.6 are indicative of confined diffusion (C.D.) or directed motion (D.M.), respectively. Almost all acidic organelles present in presynaptic terminals move by confined diffusion. Free diffusion and directed motion are present in axonal regions. **(D)** Mean $S_{MSS}$ and diffusion coefficient values for acidic organelles found in the glomerular layer (n = 82), nerve layer (n = 40) and the olfactory nerve (n = 118, seven tadpoles). Error bars indicate SEM.

followed in 35 consecutive frames or more. Endolysosomal organelles moving at maximum velocities of ~1 $\mu$m·s$^{-1}$ in shorter trajectories were consequently not considered. Such fast movements were obvious in the axonal compartment but were never observed in the presynaptic compartment (compare Videos 3 and 4). These results indicate that endolysosomes spend most of the time moving by diffusion in the cytoplasm of OSN neurons, with a representative diffusion coefficient of ~7 × 10$^{-3}$ $\mu$m$^2$·s$^{-1}$ (Fig 3D). During normal neuronal activity the endolysosomal mobility in presynaptic terminals is exclusively determined by restricted diffusion, whereas in axons, the diffusive pauses are mixed with movements associated to microtubules. Such combination of both types of mobility in axonal regions might recall the properties of the trafficking of early endosomes in cells (Zajac et al, 2013).

### SPARC activates a degradative endolysosomal pathway in axon terminals of OSNs

SPARC is a protein produced by some glial cells of the nervous system (Vincent et al, 2008) that is capable of activating a cell-autonomous program of synaptic elimination in neurons. Presynaptic terminals retract upon SPARC exposure by activating the formation of degradative endolysosomes (López-Murcia et al, 2015). Because SPARC injection in the tail of *X. tropicalis* tadpoles induces paralysis as a result of the elimination of neuromuscular junctions, we sought to obtain a loss of function effect by injecting SPARC in the olfactory bulb. Olfactory-guided behaviour was evaluated using previously described methods based on the action of amino acids as waterborne odorants (Terni et al, 2018). Local delivery of an amino acid solution through a nozzle mounted on a petri dish attracted animals to the vicinity of the odour source (Fig 4A–C). The odour-guided response was completely suppressed 5 h after injection of 50 $\mu$M recombinant SPARC in both olfactory bulbs (Fig 4D and E). Two peptidic SPARC fragments were used to confirm findings. The response to odorants was not observed when 500 $\mu$M peptide 4.2, a 20–amino acid peptide that retains most of the properties of the full-length protein (Lane & Sage, 1990), was injected (Fig 4F). In contrast, the olfactory guided response was present after 500 $\mu$M injection of peptide 2.1, a 20–amino acid peptide that does not induce synaptic elimination (López-Murcia et al, 2015). Similarly, the odour-guided behaviour was not affected in tadpoles with OSNs labelled with lysotracker deep red, thus discarding any possible toxicity induced by the dye (Fig 4G).

Transmission electron microscopy confirmed the activation of a degradative pathway as a result of SPARC microinjection. The glomerular area covered by endolysosomal organelles changed from 1.09% ± 0.3% to 3.1% ± 0.5% or 3.9% ± 1% after injection of p4.2 or SPARC, respectively ($P$ < 0.001, Kruskal–Wallis test followed by Dunn's multiple comparisons test). The activation of an endolysosomal pathway was characterized by the presence of lysosomes (Fig 5A), late endosomes (Fig 5B and C) and endolysosomes (Fig 5D and E) in axon terminals of tadpoles injected with SPARC or peptide p4.2. Double membrane autophagic intermediates were also obvious (Fig 5F and G). The wide distribution of these organelles throughout the glomerular layer of the olfactory bulb suggested an ongoing cell-autonomous degradative mechanism of synaptic elements, as previously reported in vitro (López-Murcia et al, 2015). Retraction bulbs,

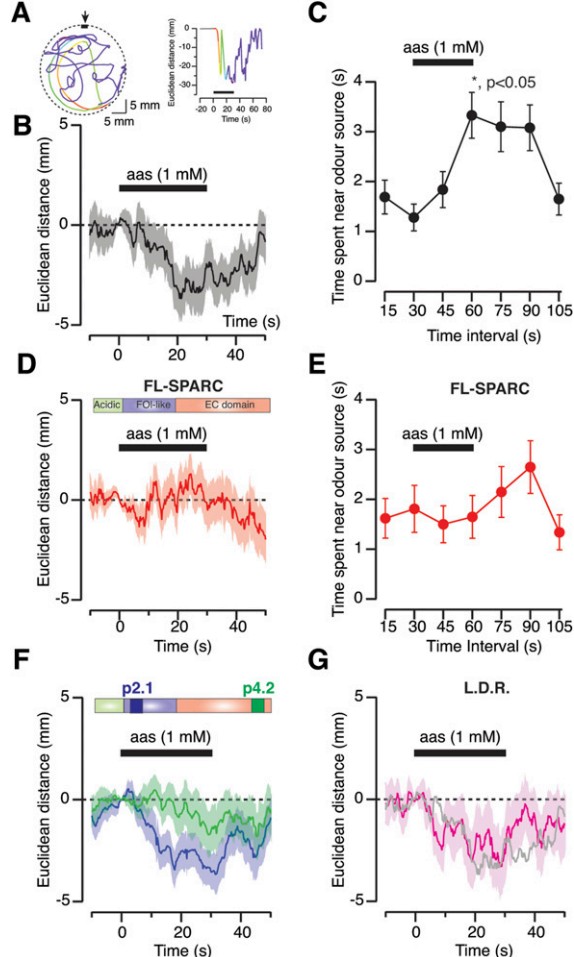

**Figure 4. Microinjection of secreted protein acidic and rich in cysteine (SPARC) in the olfactory bulb disrupts olfactory-guided behaviour.**
**(A)** *Xenopus tropicalis* tadpoles are attracted by local delivery of a 1 mM mixture of five different amino acids (methionine, leucine, histidine, arginine, and lysine), which act as waterborne odorants. Changes in the Euclidean distance found between the animal and the nozzle (arrow) delivering the odorants set the bases for quantifying olfactory-guided behaviour. For analysis, all tadpoles are considered to be 0 mm away from the odour source just before exposure to odorants. Negative distance values show attraction, whereas positive values indicate repulsion. **(B)** Average response of control tadpoles to local delivery of the amino acid solution (n = 72). Solid line and shadowed area indicate mean Euclidean distance and SEM, respectively. **(C)** Quantification of the time spent in the vicinity of the odour source (see the Materials and Methods section for details). The attraction for waterborne odorants is transient and occurs from 15 to 30 s after amino acids start to flow into the well (*$P$ < 0.05, Kruskal–Wallis $t$ test followed by Dunn's multiple comparisons test). Dots indicate mean ± SEM (n = 72). **(D)** Average olfactory-guided response of tadpoles 5 h after injection of 50 $\mu$M full-length recombinant SPARC in the olfactory bulb (n = 78). Solid line and shadowed area indicate mean ± SEM changes in the Euclidean distance separating the animal and the nozzle delivering the amino acids. **(E)** Tadpoles injected with full-length SPARC are not attracted by waterborne odorants. Dots indicate mean ± SEM (n = 78). **(F)** Average olfactory guided response of tadpoles 5 h after injection of 500 $\mu$M peptide p4.2 (n = 51) or 500 $\mu$M peptide p2.1 (n = 54) in the olfactory bulb. Both peptides contain 20 amino acids found in the illustrated regions of SPARC. Peptide 4.2 displays an analogous activity to the full-length protein. Peptide 2.1 is inactive and is used as a control of injection. Solid line and shadowed area show mean ± SEM changes in Euclidean distance. **(G)** Labelling of olfactory sensory neurons with lysotracker deep red did not alter olfactory-guided behaviour (n = 58). Solid line and shadowed area indicate mean ± SEM changes in Euclidean distance. The grey trace indicates the control response (from B) for comparison purposes.

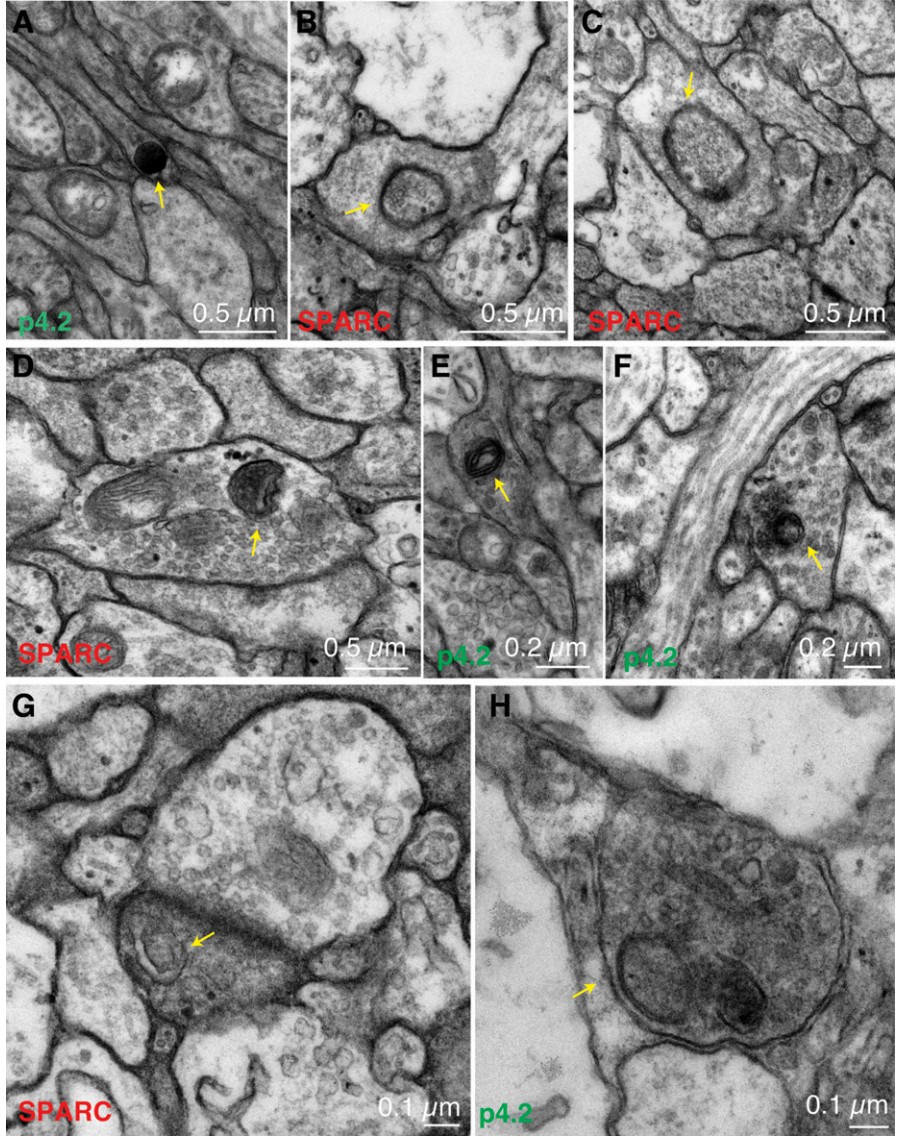

**Figure 5. Presence of multiple organelles of the endolysosomal system in axon terminals of olfactory sensory neurons during synaptic remodelling.**
**(A, B, C, D, E, F, G)** Presynaptic terminals forming olfactory glomeruli were inspected for the presence of components of the endolysosomal system upon exposure to full-length secreted protein acidic and rich in cysteine (SPARC) or its active peptidic fragment p4.2. Lysosome (A, arrow), late endosomes (B, C, arrows), endolysosomes (D, E, arrows), and autophagic intermediates (F, G, arrows). **(H)** Retraction bulb (arrow) suggesting an ongoing elimination of synaptic contacts.

which contained multiple degradative organelles, were also observed (Fig 5H). They were considered an evidence of synapse loss (Hill 2017) and were never observed in control tadpoles or animals injected with peptide p2.1. These observations are in agreement with previous studies demonstrating the ability of SPARC to activate autophagy in cells of the nervous system (Bhoopathi et al, 2010) and support that the induction of a degradative synaptic pathway upon injection of both, peptide p4.2 and SPARC, was likely responsible for the loss of olfactory-guided behaviour.

### SPARC induces the remodelling of the F-actin cytoskeleton and increases the diffusion of presynaptic endolysosomes

Numerous F-actin patches were visualized by STED stimulated emission depletion (STED) microscopy in olfactory glomeruli stained with phalloidin-647A (Fig 6A). Intensity profiles drawn along the discrete fluorescent spots showed a peak of F-actin concentration that gradually decayed to cover a surface ranging from 0.04 to 1.09 $\mu m^2$. This characteristic structure of F-actin patches was altered upon exposure of the olfactory bulb to SPARC as they became wider and less intense (Fig 6B). The changes in the morphology of F-actin patches could reflect the biological activity of SPARC, which is based on its ability to reorganize the actin cytoskeleton (Murphy-Ullrich & Sage, 2014). The average intensity of the profiles revealed a decrease of the maximal F-actin concentration and a redistribution of F-actin within the patch (Fig 6C and D). The 15% increase in width (Fig 6E) was associated to a 35% increase in the patch area (Fig 6F). Taking into account the key role of F-actin patches in the maintenance of neuronal morphology (Lavoie-Cardinal et al, 2020), as well as the ability of SPARC to trigger synaptic elimination after F-actin remodelling (Velasco & Llobet, 2020), it is possible that the alterations caused by SPARC on glomerular F-actin are a hallmark for the activation of a degradative synaptic pathway.

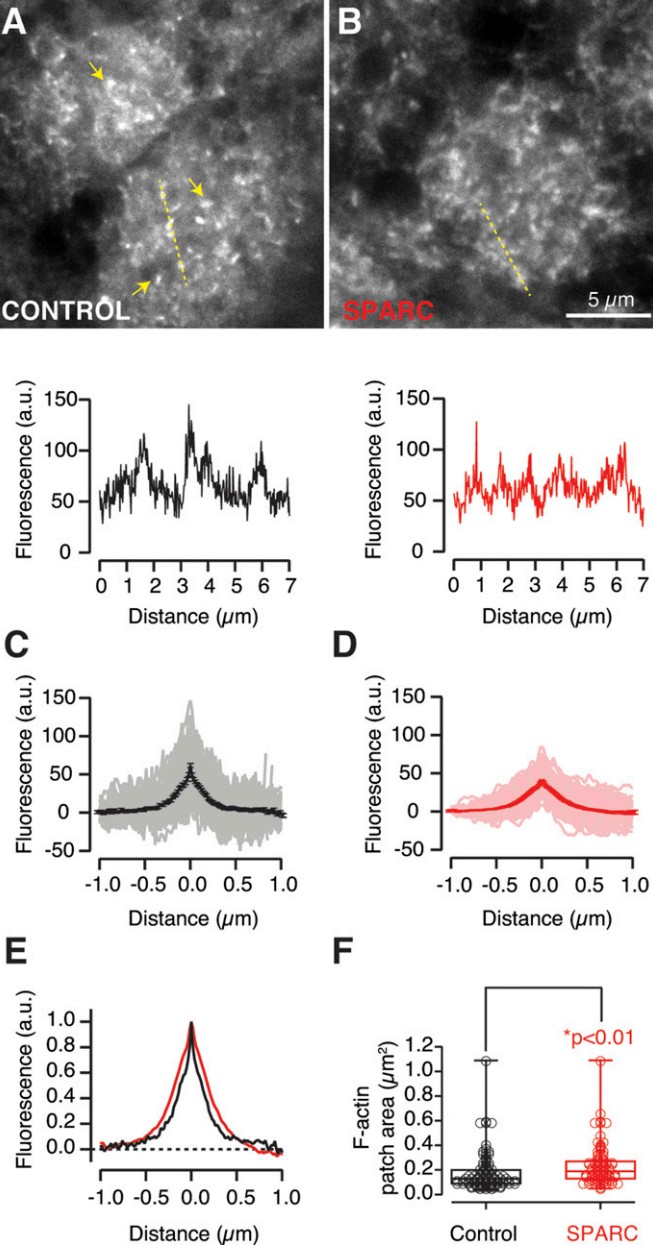

**Figure 6. Secreted protein acidic and rich in cysteine (SPARC) alters the structure of F-actin patches present in olfactory glomeruli.**
**(A)** F-actin patches (arrows) were present in olfactory glomeruli (up). Plot of Intensity profiles (dotted yellow line) used to estimate the width and maximum fluorescence of the structures identified (down). **(B)** SPARC disrupted the appearance of F-actin patches by increasing their area and reducing their fluorescence peak. **(C, D)** Average intensity profiles from F-actin patches obtained in control condition (C, n = 92) or in tadpoles injected with 50 µM SPARC (D, n = 137). Solid lines and error bars indicate mean ± SEM obtained from the individual traces shown. **(E)** Normalized average intensity profiles showing an increase in the width of F-actin patches in tadpoles injected with SPARC (red) compared with control animals (black). **(F)** Comparison of the area covered by F-actin patches in the olfactory glomeruli of control (black, n = 104) and SPARC injected tadpoles (red, n = 95). Box plots show the median (horizontal line), 25–75% quartiles (boxes), and ranges (whiskers) of the area of F-actin patches. The difference between the two groups was established by the *t* test.

We next evaluated the effect of latrunculin-A to investigate the direct implication of the presynaptic F-actin cytoskeleton on the mobility of endolysosomes (Fig 7A). The diffusion coefficient of endolysosomes increased twofold as a result of depolymerization of F-actin (Fig 7B). Interestingly, the effect of peptide p4.2 and SPARC on the mobility of lysotracker labelled organelles was similar, driving also to an approximately twofold increase in the diffusion coefficient (Fig 7B and C). Therefore, the disruption or alteration of the presynaptic F-actin cytoskeleton caused a faster confined diffusion of endolysosomes. Injection of the inactive fragment of SPARC, peptide p2.1, did not modify mobility (Fig 7C) and $S_{MSS}$ remained unaltered in all experimental groups evaluated (Fig 7B).

The change in diffusivity must be interpreted in the context of the particular morphology of OSN axon terminals, which do not form a classical presynaptic bouton. They define a long, intricate, tortuous presynaptic compartment that typically measures 20–50 µm and is filled with glutamatergic synaptic vesicles (Nezlin et al, 2003; Hassenklöver & Manzini, 2013; Terni et al, 2017). A rough estimation of the time (t) required for an endolysosome to move a certain distance (r) in the three dimensions of the axon terminal can be obtained from the diffusion coefficient, using the expression t = r2/6D (Berg, 1993). In a control animal, it would take ~50 min to travel along 10 µm of the presynaptic compartment, whereas during synaptic remodelling, this time scale would be reduced to about its half. Either way, endolysosomes would always remain during tens of minutes in the presynaptic compartment.

Presynaptic endolysosomal organelles moved slowly, at an average velocity of ~0.1 $\mu\text{m·s}^{-1}$ (Fig 7D), which is about an order of magnitude smaller than in the cytoplasm of cultured cells (Bandyopadhyay et al, 2014). Upon remodelling of synaptic F-actin with latrunculin-A or SPARC, the average velocity raised up to ~0.2 $\mu\text{m·s}^{-1}$, in agreement with the change observed in the diffusion coefficient (Fig 7B). This increase was far from the velocities >1 $\mu\text{m·s}^{-1}$ observed for the mobility of organelles along microtubules (Zajac et al, 2013) and confirmed that diffusion determines endolysosomal mobility in the presynaptic compartment. However, from 10% to 20% of particles transiently reached maximum velocities ≥1 $\mu\text{m·s}^{-1}$ in all experimental groups investigated (Fig 7D), which might reflect the coexistence of a default movement by diffusion with short-lived displacements along microtubules.

The residence time of endolysosomes found in axon terminals defined by diffusion could set the relevance of local proteolytic activity (see Fig 2A), against their ability to undergo fast retrograde transportation to the cell body (Encalada & Goldstein, 2014). The increase in the local degradative activity within the presynaptic compartment (Fig 5) is supported by an apparent enlargement of endolysosomal organelles. The average radius of 0.28 µm found in control conditions (n = 60, 8 tadpoles) increased to 0.41 µm in tadpoles injected with SPARC (n = 72, 10 tadpoles, Fig 8A). The presence of larger acidic organelles could be the hallmark of a change from homeostatic to net degradative activity, reminiscent of the ability of endolysosomes to expand their volume during phagocyte activation (Hipolito et al, 2019).

Assuming the presynaptic cytosol behaves as a simple solution with greater viscosity than water (Wojcieszyn et al, 1981), we evaluated

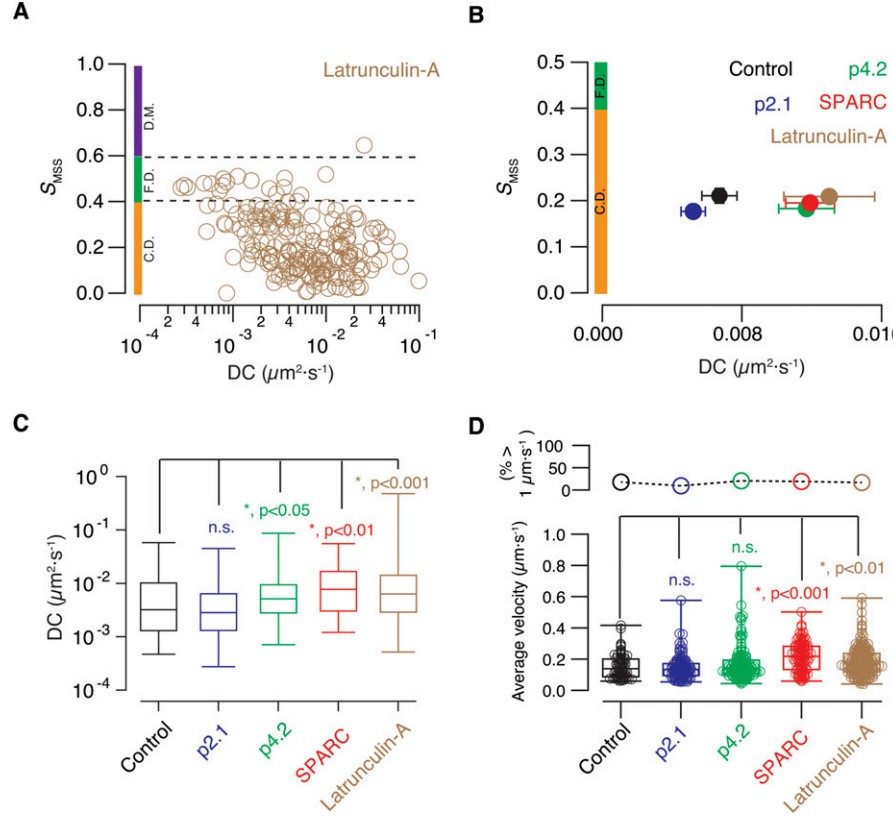

**Figure 7. The diffusion coefficient of endolysosomes found in axon terminals is increased during synaptic remodelling.**
**(A)** Acute effect of 50 $\mu$M latrunculin-A on the mobility of acidic organelles present in presynaptic terminals. Plot of the slope of the moment scaling spectrum ($S_{MSS}$) against the diffusion coefficient (DC) of particles labelled with lysotracker deep red. $S_{MSS}$ values found between 0.4 and 0.6 are associated to free diffusion (F.D., dotted lines), whereas $S_{MSS}$ values <0.4 and >0.6 are indicative of confined diffusion (C.D.) and directed motion (D.M.), respectively. **(B)** Relationship between the $S_{MSS}$ and the DC of acidic organelles found in axon terminals of olfactory sensory neurons in five different experimental conditions. Dots indicate mean ± SEM obtained in the indicated groups. Control (n = 75, 8 tadpoles), p2.1 (n = 67, 8 tadpoles), p4.2 (n = 143, 6 tadpoles), full-length secreted protein acidic and rich in cysteine (SPARC, n = 72, 10 tadpoles), and latrunculin-A (n = 209, 14 tadpoles). Each dot indicates mean ± SEM. Colored bars indicate the ranges of $S_{MSS}$ values associated to confined diffusion (C.D.) and free diffusion (F.D.). **(C)** Box plots show the median (horizontal line), 25–75% quartiles (boxes), and ranges (whiskers) of DCs found for presynaptic acidic organelles in the five experimental groups showed in [B]. Statistical differences were established using the Kruskal–Wallis test followed by Dunn's multiple comparisons test. **(D)** Box plots show the median (horizontal line), 25–75% quartiles (boxes), and ranges (whiskers) of the average velocity of presynaptic acidic organelles found in the five experimental groups showed in [B]. Statistical differences were established using Kruskal–Wallis test followed by Dunn's multiple comparisons test. The percentage of particles reaching velocities >1 $\mu$m·s$^{-1}$ in each experimental group is indicated in the upper plot.

how changes in endolysosomal size were related to diffusivity using the Stokes–Einstein Equation (1),

$$D = \frac{kT}{6\pi\eta r};\qquad(1)$$

where D is the diffusion coefficient, k is the Boltzmann's constant, T is absolute temperature, $\eta$ is viscosity, and r is the radius of the particle labelled with lysotracker deep red. Fitting of data to Equation (1) reported an apparent viscosity of the presynaptic cytosol of 0.15 Pa·s (Fig 8B). This value implied that the viscosity in the presynaptic compartment was about 20 times higher than in the cell cytoplasm estimated in photobleaching experiments also using Equation (1) (Lang et al, 1986). The remodelling of synaptic connectivity induced by SPARC altered the relationship between diffusivity and organelle size and was consistent with a decrease in cytosolic viscosity to 0.066 Pa·s (Fig 8C). The cause could be the disruption of the organization F-actin patches by SPARC (Fig 6). This possibility was considered in the light of previous findings proposing an important contribution of the actin cytoskeleton to viscosity (Hou et al, 1990) and it was investigated in tadpoles injected with 50 $\mu$M latrunculin-A in the olfactory bulb. F-actin depolymerization led to an increase in the size and mobility of endolysosomal organelles similar to those induced by SPARC, which was related to an average decrease of cytosolic viscosity to 0.085 Pa·s (Fig 8D and E). The properties of the cytosol were apparently not affected in the experimental group using

peptide 2.1 as a control of microinjection (Fig 8F). These results indicate that pathways driving to a disruption or a remodelling of the presynaptic F-actin cytoskeleton could have a direct effect on the viscosity of the cytosol and, consequently, the endolysosomes generated during a process of synaptic remodelling or elimination would be particularly affected. For instance, endolysosomal organelles measuring >1 $\mu$m diameter could transition from being essentially stalled (Fig 8B, DC ~0.001 $\mu$m$^2$·s$^{-1}$) to slowly diffuse (Fig 8C, DC ~0.005 $\mu$m$^2$·s$^{-1}$).

Latrunculin-A injection in the olfactory bulb also altered olfactory-guided behaviour but, the effect was qualitatively different from SPARC. Latrunculin-A delayed the response to odorants by ~12 s (Fig 8G), whereas SPARC actually supressed it (Fig 4B). SPARC and latrunculin alter the F-actin cytoskeleton, which explains a common ability to increase the mobility of endolysosomes; however, their synaptic actions are different: SPARC eliminates synapses (López-Murcia et al, 2015), but latrunculin-A impairs neurotransmission (Bleckert et al, 2012) and synaptic contacts persist (Morales et al, 2000).

## Retrograde transport of endolysosomes is not modified during synaptic remodelling

We next evaluated if the increase in diffusivity of endolysosomes during synaptic remodelling was associated to their transfer from presynaptic terminals to axons by investigating retrograde and anterograde axonal transport. In control conditions the net movement of acidic organelles

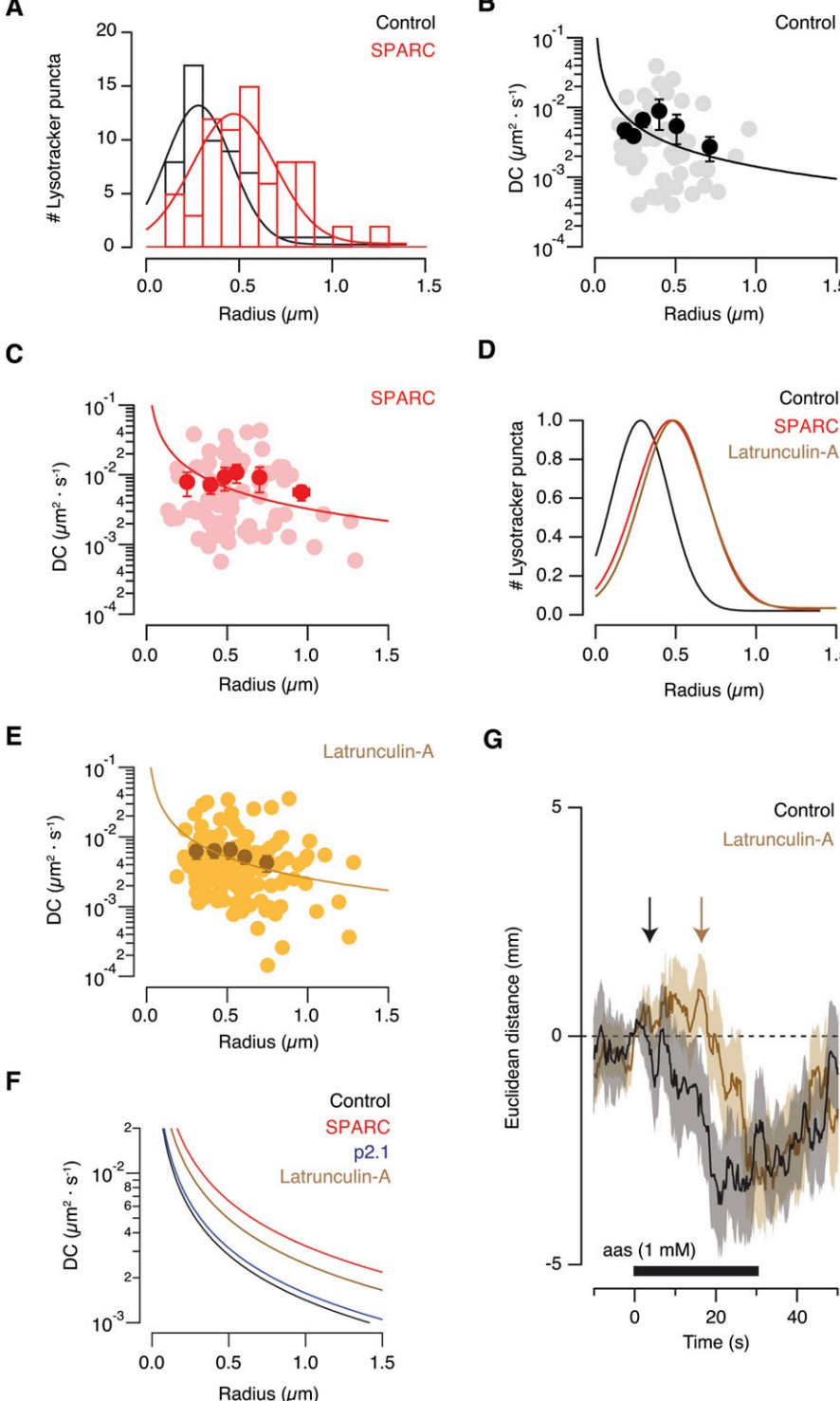

**Figure 8.  The increase in the mobility of presynaptic endolysosomes is associated to changes in the viscosity of the cytosol.**
**(A)** The radius of presynaptic endolysosomes increased when full-length secreted protein acidic and rich in cysteine (SPARC) triggered the remodelling of synaptic connectivity. Control (n = 60, eight tadpoles) and SPARC-injected tadpoles (n = 72, ten tadpoles) **(B, C)**. The diffusion coefficient decreased as a function of the radius of puncta labelled with lysotracker deep red. Fitting of individual values to the Stokes–Einstein equation revealed that the viscosity ($\eta$) of 0.15 Pa·s found in axon terminals decreased to 0.066 Pa·s after the glomerular layer was exposed to SPARC. Bins indicate mean ± SEM. and dots show individual values. **(D)** Normalized Gaussian fits show that latrunculin-A and SPARC increased the size of acidic organelles compared to control conditions. **(E)** Latrunculin-A reduced the viscosity of the presynaptic cytosol to 0.085 Pa·s. Bins indicate mean ± SEM and dots show individual values. **(F)** Microinjection of the inactive SPARC derived peptide, p2.1, did not alter the viscosity of the presynaptic compartment ($\eta$ = 0.14 Pa·s). Fits obtained in (B, C, E) are included for comparison purposes. **(G)** Average olfactory guided response (n = 56) of tadpoles injected with 50 $\mu$M latrunculin-A to local delivery of a 1 mM mixture of 5 different amino acids (methionine, leucine, histidine, arginine, and lysine). Solid line and shadowed area indicate mean Euclidean distance and SEM, respectively. The control response of non-injected animals is displayed for comparison purposes. Arrows show the time of initiation of the movement towards the odour source. Notice the delay present in tadpoles injected with latrunculin-A (time between arrows).

in the olfactory nerve was balanced, albeit there was a slight shift towards retrograde transportation. In general terms and, as indicated in the example of Fig 9A, there were as many endolysosomes moving anterogradely as retrogradely. This behaviour was consistent in all observed trajectories (n = 7 tadpoles) and was maintained when control

injections were carried out using peptide p2.1 (n = 7 tadpoles, Fig 9B). The balance between retrograde and anterograde transport of acidic organelles was also not modified during conditions inducing synaptic remodelling. Neither peptide p4.2 nor SPARC altered normal axonal transport, so that no differences were found among all experimental

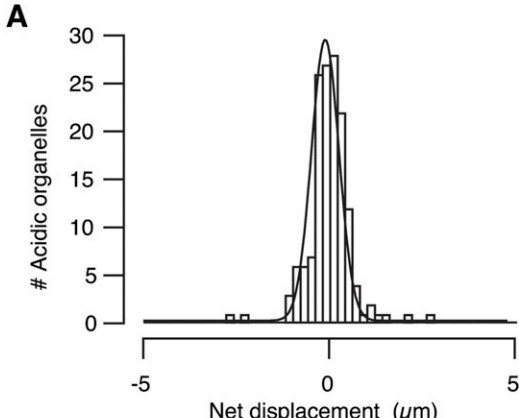

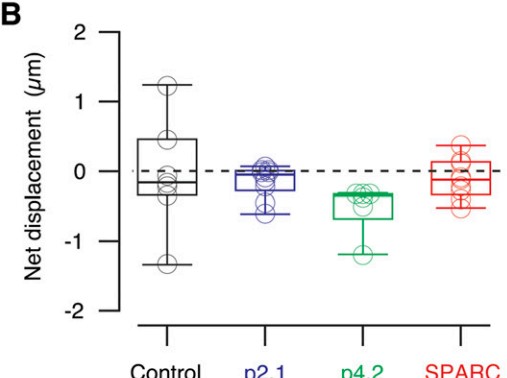

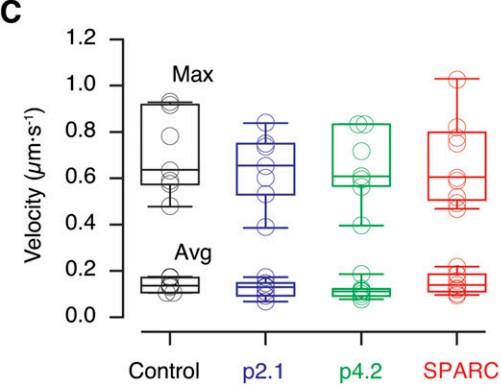

**Figure 9. Axonal transport of endolysosomes is not affected by synaptic remodelling.**

**(A)** Histogram of net displacements of all acidic organelles found in the olfactory nerve (ON) of a control tadpole occurring in a time interval of 200 s (n = 152). Negative values indicate retrograde transport and positive values indicate anterograde transport. A Gaussian fit was used to obtain the mean value of particle displacements, which in this example was of −0.105 μm. This distance was considered a quantitative indicator of the balance between retrograde and anterograde transport of endolysosomes in the ON and was representative of a given tadpole. **(B)** No differences (one-way ANOVA analysis followed by Dunnett's multiple comparisons test) were found among the mean, net representative displacements of acidic organelles present in the ONs in the indicated experimental conditions. Control (n = 7 tadpoles), p2.1 (n = 7 tadpoles), p4.2 (n = 7 tadpoles), full-length secreted protein acidic and rich in cysteine (SPARC, n = 9 tadpoles). **(C)** No differences (one-way ANOVA analysis followed by Dunnett's multiple comparisons test) were found in the average velocity nor the maximum velocity of endolysosomes present in the ONs in the indicated experimental conditions. Control (n = 7 tadpoles), p2.1 (n = 7 tadpoles), p4.2 (n = 7 tadpoles), and SPARC (n = 9 tadpoles).

groups evaluated ($P$ = 0.27, one-way ANOVA analysis, Fig 9B). These results were consistent with a compartmentalized action of presynaptic endolysosomes to support a local degradative activity; however, they could also be explained by an increase in the traffic of endolysosomal organelles without altering the balance between retrograde and anterograde transport. The representative average and maximum velocities of acidic organelles found in the olfactory nerves of animals investigated did not change upon injection of peptide p4.2 or SPARC ($P$ = 0.30 and $P$ = 0.92, respectively, one-way ANOVA analysis, Fig 9C). The absence of any obvious change in the velocity or directionality of axonal endolysosomes evidenced that the increase in the degradative activity in the presynaptic compartment was not associated to an enhancement of their retrograde transportation. Therefore, acidic organelles present in the presynaptic compartment contributed locally to the remodelling of synaptic contacts triggered by SPARC.

## Discussion

Endolysosomes are present in axon terminals of *X. tropicalis* OSN neurons, which implies the presence of an ongoing, homeostatic degradative activity (Azarnia Tehran et al, 2018). Here, we show in in vivo observations that the activation of a cell-autonomous mechanism of synaptic degradation mediated by endolysosomes is related to an increase in the diffusion of these acidic organelles within the presynaptic compartment.

Typical diameters of lysosomes and endosomes found at their different maturation stages range from 100 nm to 1 μm (Klumperman & Raposo, 2014), thus meaning their volume can be two to three orders of magnitude larger than synaptic vesicles (Takamori et al, 2006). In small hippocampal boutons, the diffusion coefficients of synaptic vesicles range from $5 \times 10^{-5}$ to $1 \times 10^{-3}$ $\mu m^2 \cdot s^{-1}$ (Jordan et al, 2005; Guillaud et al, 2017) and change as a function of activity (Forte et al, 2017). Assuming the exclusive presence of random, diffusive motion, there is an inverse relation between the diffusivity and organelle radius (Equation [1], [Bandyopadhyay et al, 2014], see also Brangwynne et al [2008]). The mobility of endolysosomes within the presynaptic terminal is thus expected to be slower than that of synaptic vesicles; however, the diffusion coefficient of $6.7 \times 10^{-3}$ $\mu m^2 \cdot s^{-1}$ found for acidic presynaptic endolysosomal organelles is well within the range of synaptic vesicles. The explanation can be found in the completely different molecular composition of endolysosomes and synaptic vesicles. Endolysosomes lack synapsins, which are key to restrict the mobility of synaptic vesicles and promote their clustering (Orenbuch et al, 2012; Pechstein et al, 2020). Actually, glutamatergic synaptic vesicles present in retinal bipolar cell terminals, a type of ribbon synapse that lacks synapsin isoforms (Mandell et al, 1990), display a diffusion coefficient of 0.015 $\mu m^2 \cdot s^{-1}$ (Holt et al, 2004). Therefore, small synaptic vesicles that lack synapsins do not have the ability of establishing clusters and diffuse faster than presynaptic endolysosomal organelles. These evidences support that the variable molecular composition of the organelles of the endolysosomal system (Luzio et al, 2014) are, together with size, determinants of their diffusion in the presynaptic compartment. Our observations open the intriguing possibility that the molecular events driving to endolysosome

formation could be orchestrated to regulate size as well as membrane composition with the aim of altering the diffusivity of the organelle.

A default mobility based on the diffusion of endolysosomal organelles could favour the establishment of intracellular interactions. An example could be the fusion with autophagosomes that are formed in synapses and acidify on their journey to the soma (Maday et al, 2012). Fusion could occur either in the presynaptic compartment or even along the axon during diffusive pauses, which alternate with active retrograde transport. Thus, the importance of diffusion to favour organelle interactions could depend on the individual morphology of presynaptic terminals. The presynaptic compartment of OSNs forms a glomerulus that measures tens of microns and contains a large cytoplasmic pool of glutamatergic synaptic vesicles (Nezlin et al, 2003; Terni et al, 2017). It is thus conceivable that endolysosomes reside longer time intervals in OSN terminals than in small presynaptic boutons measuring ~1 $\mu$m diameter and containing ~200 vesicles (Schikorski & Stevens, 1997).

Can presynaptic terminals control the diffusion of endolysosomal organelles? SPARC, a molecule secreted by glia that triggers a cell-autonomous mechanism of synaptic elimination in cholinergic axon terminals (López-Murcia et al, 2015) suppressed olfactory-guided behaviour and increased the presence of organelles of the endolysosomal pathway, as well as their diffusion within the presynaptic compartment of OSNs. The action of SPARC is thus not restricted to excitatory synapses of the peripheral nervous system and can be extended to glutamatergic synapses. The activation of a degradative endolysosomal pathway in axon terminals was not characterized by the retrograde transportation of digestive organelles to the cell body after their formation but, by an increase in their local mobility. An overall increase in diffusion implies that a large organelle, for example, a late endosome, acquires a mobility comparable to a smaller organelle. The chance of fusing with a lysosome is therefore enhanced and the generated endolysosome can take advantage of a greater mobility to continue shaping its composition by establishing interactions with other organelles. The endolysosome can finally exit the presynaptic terminal to start its journey to the soma while continuing with maturation (Ferguson, 2018). These results highlight the importance of all the molecular mechanisms regulating the size of lysosomes (reviewed in de Araujo et al [2020]) and prompt to consider them as determinants of the maturation time of endolysosomes in the presynaptic compartment. In a context of mobility exclusively determined by diffusion, very large degradative organelles could not even be able to move and become stalled. The inability to leave the presynaptic compartment for undergoing retrograde transportation to the cell body could make necessary to invoke the participation of other pathways. An example could be axosome shedding, a process that occurs during postnatal elimination of neuromuscular junctions consisting in the digestion of large accumulations of presynaptic elements by terminal Schwann cells (Bishop et al, 2004).

Two main evidences support that the mobility of lysosomes is controlled by F-actin in dendrites: they are preferentially located at F-actin patches (van Bommel et al, 2019) and become more mobile after disruption of F-actin with latrunculin-A (Goo et al, 2017; van Bommel et al, 2019). Our results support that F-actin might play an analogous role in the presynaptic compartment. The disruption of the structure of F-actin patches present in olfactory glomeruli by SPARC had a similar effect to latrunculin-A on the mobility of endolysosomes. This finding could be explained by a decrease in the viscosity of the presynaptic cytosol, assuming organelle motion followed exclusively a random walk. Taking into account that F-actin filaments are key determinants of cytoplasmic viscosity (Hou et al, 1990) and presynaptic structure (Sankaranarayanan et al, 2003), it is thus possible that the ability of SPARC to reorganize the F-actin cytoskeleton (Murphy-Ullrich et al, 1995) had a direct effect on the physical properties of the presynaptic cytosol. We cannot discriminate if the effect of SPARC were general and affected to the entire meshwork of F-actin filaments present in the presynaptic compartment, or in contrast, were local. The first possibility could be supported by the similar action of SPARC and latrunculin-A, whereas the second situation could be considered by drawing an analogy with the dendritic compartment (van Bommel et al, 2019). A targeted decrease of F-actin concentration restricted to patches could reflect an impairment of the presynaptic compartment to position endolysosomes and also explain an increase in motion.

Axons contain periodic F-actin repeats (D'Este et al, 2015) and in certain regions, such as the initial segment, the periodic pattern of actin, spectrin, and ankyrin confines membrane proteins (Albrecht et al, 2016). According to our results the organized F-actin cytoskeleton could also be determinant of the diffusion of presynaptic organelles. In this scenario, synaptic maturation could ultimately contribute to the mobility of endolysosomes because synapses gradually become more stable throughout development. An ordered membrane cytoskeleton could confine endolysosomal organelles and consequently limit their degradative activity by constraining the establishment of interactions with other membranous compartments. Such regulated, limited mobility could favour a homeostatic role. In contrast, a less structured presynaptic membrane cytoskeleton could decrease the restricted motion of organelles and facilitate the interactions required to activate a local degradative endolysosomal pathway.

# Materials and Methods

### Animals

Ethical procedures were approved by the regional government (Generalitat de Catalunya, experimental procedure #10753). *X. tropicalis* were housed and raised according to the standard protocols of the animal facilities of the University of Barcelona. Larvae were obtained by natural mating and kept in water tanks at 25°C. Water conductivity was adjusted to 700 $\mu$s·cm$^{-1}$ and pH 7.5. Tadpoles at stage 47–50 of the Nieuwkoop–Faber criteria were used for the experiments (Nieuwkoop & Faber, 1956). The *X. tropicalis* transgenic line *zHB9::GFP* (RRID:NXR_1.0111) was used for the visualization of a discrete population of OSNs (Terni et al, 2017). The labelling of the olfactory epithelium in this line is considered ectopic because the zHB9 promoter drives GFP expression in motor

neurons (Flanagan-Steet et al, 2005). It is, however, remarkable that a comparable expression pattern is observed in HB9 transgenic mice (Nakano et al, 2005). The labelling of OSNs was constant in all animals of the *zHB9::GFP* transgenic line.

## In vivo imaging of endolysosomal organelles

The two nasal cavities of *zHB9::GFP* tadpoles anesthetized with 0.02% MS-222 were filled with 0.15–0.3 $\mu$l of 1 mM lysotracker deep red (Reference: L12492; Thermo Fisher Scientific). In a set of experiments, lysotracker deep red was injected together with magic red cathepsin-B (Bio-Rad Laboratories Ltd). A microinjector (Picospritzer III; Parker Hanifin corporation) delivered a puff of the dye (30 $\psi$, 6 ms) using ~1-$\mu$m tip pipettes fabricated from borosilicate glass capillaries (Reference: BF120-94-10; Sutter Instrument). Animals recovered from anaesthesia in 2-liter tanks filled with tadpole water. The remodelling of olfactory bulb synapses was activated by the injection of 40–60 nl of 50 $\mu$M recombinant human SPARC protein (R&D Systems) or 500 $\mu$M p4.2, a peptidic sequence of 20–amino acids that mimics the action of the full-length protein (López-Murcia et al, 2015; Velasco & Llobet, 2020). 500 $\mu$M peptide p2.1, a 20–amino acid sequence located in the follistatin-like domain of SPARC, was used as control of injection. A stock solution of 2 mM latrunculin-A (Cat. Number: 3973; Tocris Bioscience) was prepared in DMSO and injected to a final concentration of 50 $\mu$M in the indicated experiments. Recombinant SPARC, SPARC-derived peptides, and latrunculin-A were dissolved in *Xenopus* Ringer, which contained (in mM): 5 Hepes, 100 NaCl, 2 KCl, and 1 MgSO$_4$ (240 mOsm/kg, pH = 7.8). Microinjections were carried out at a pressure of 30 $\psi$ using calibrated borosilicate glass pipettes. Typical injection times were 5–10 ms. Imaging sessions started 5 h after injection of peptides p2.1, p4.2 and SPARC, a time window required to induce synaptic remodelling (López-Murcia et al, 2015; Velasco & Llobet, 2020). The action of latrunculin-A was evaluated 30–45 min after injection.

Imaging sessions started by anesthetising tadpoles in water containing 0.02% MS-222 to remove the skin covering the olfactory bulb. The goal was to eliminate melanocytes that could interfere with imaging. Anesthetized animals were placed dorsally on a 25 mm diameter coverslip (#1.5 thickness, Reference: 631-0172; VWR International GmbH) and embedded in a 1% solution of low melting point agarose (Reference: A9414-10G; Sigma-Aldrich). Imaging experiments were carried out 18–24 h after application of lysotracker deep red in a Zeiss LSM 880 confocal microscope (Carl Zeiss AG). After the identification of GFP expression in olfactory glomeruli and the presence of discrete puncta along OSNs, tadpole viability was certified by observing the blood flow in vessels running along the olfactory nerve (Video 2). An imaging session lasted 10–15 min.

Confocal sections were acquired using a 63× glycerol immersion objective LD LCI Plan-Apochromat (1.2 N.A.). GFP and lysotracker deep red were excited at 488 and 633 nm, respectively. Lysotracker deep red fluorescence was collected at 700 nm using a GaAsp detector. Cathepsin-B magic red was excited at 555 nm and the emission spectra were limited to avoid the excitation of lysotracker deep red. Tracking of labelled acidic organelles was carried out in 1,024 × 1,024 pixel confocal sections at a resolution of 7.59 pixels·$\mu$m$^{-1}$ (voxel size 0.132 × 0.132 × 0.499 $\mu$m$^3$). Experiments started by obtaining a Z-stack to observe the morphology of OSN axonal processes labelled with

GFP that formed a glomerulus, as well as the distribution of acidic organelles stained with lysotracker deep red. A time series of a single 2D confocal section was collected at 0.8 Hz for 4 min. The selected region was situated dorsally to maximize light transmission. The procedure was repeated in a central region of an olfactory nerve upon finalizing imaging at the glomerular level.

## Image analysis

Image analysis was carried out using Image J and Fiji. The drift in the X-Y plane was corrected using the StackReg plugin (Thévenaz et al, 1998). MosaicSuite plugins (Sbalzarini & Koumoutsakos, 2005) were used to select image trajectories, as well as to calculate the slope of the moment scaling spectrum ($S_{MSS}$) and the diffusion coefficient of identified acidic organelles. Only trajectories of single particles that were tracked for >45 s were considered. Particle positions were imported in Igor Pro 8 (WaveMetrics) to calculate the distance moved, average velocity and maximum velocity using custom-made macros. Rotated images displaying olfactory nerves along the x-axis were used to quantify retrograde and anterograde transport of acidic organelles. The trajectories identified with the TrackMate plugin (Tinevez et al, 2017) were imported into Igor Pro 8 and analysed with custom-made macros. Particle size was estimated using line profiles drawn along individual organelles. The radius was calculated by dividing by two the full width at half maximum obtained from Gaussian fits.

## Assessment of olfactory-guided behaviour

The assay of an olfactory-guided response was performed as previously described (Terni et al, 2018). Free swimming tadpoles were placed in a six-well dish where each well contained 10 ml of *Xenopus* water and a single animal. Tadpoles adapted to the new environment during 3–5 min before the delivery of odorants, which consisted in a mixture of five different amino acids (methionine, leucine, histidine, arginine, and lysine) acting as a broad-range stimulus of OSNs (Manzini et al, 2007). The solution of waterborne odorants was prepared at a final concentration of 1 mM in *Xenopus* water (conductance: 700 $\mu$S·cm$^{-1}$; pH: 7.8). The amino acid solution was kept in in an elevated reservoir, connected to a six-line manifold using propylene tubing. Upon opening a clamp, individual perfusion inlets allowed the delivery of the solution of amino acids for 30 s. Animal swimming was continuously recorded at 6 Hz using a digital MRC5 camera (Carl Zeiss AG). Movies were imported in Image J and analysed with the MTrackJ and Wrmtrck plugins (Meijering et al, 2012; Nussbaum-Krammer et al, 2015). Individual trajectories were imported in Igor Pro 8.0 for analysis (Terni et al, 2018). Behavioural assays were carried out 5 h after injection of peptides p2.1, p4.2, SPARC, and latrunculin-A.

## Histological procedures

Tadpoles were fixed for immunohistochemistry in 4% PFA during 4–7 d at 4°C and immersed in 30% sucrose. Animals were next embedded in O.C.T. freezing medium (Tissue-Tek; Sakura Finetek), snap-frozen in isopentane in a Bright Clini-RF rapid freezer and stored at −80°C until use. Horizontal sections (15 $\mu$m thick) were obtained using a cryostat (Reichert-Jung; Leica) and mounted on

superfrost plus slides (VWR International GmbH). For immunofluorescent staining the sections were blocked for 2 h at RT with PBS solution containing 0.2% Triton X-100 and 10% normal goat serum. They were next incubated in a moist chamber overnight at 4°C in PBS with 0.2% Triton X-100 and 2% normal goat serum containing anti-synaptophysin (mouse monoclonal, 101011, 1:200, RRID: AB-887824; Synaptic Systems) and LAMP-1 (rabbit polyclonal, 24170, 1:500, RRID: AB-775978; Abcam). After three washes with PBS, sections were incubated with appropriate secondary antibodies followed by DAPI staining (1:10,000) and mounted with Fluoromount (Sigma-Aldrich). Horizontal sections of 10-µm thick were prepared for visualization by STED microscopy. Incubation with 0.165 µM phalloidin-Atto647N (Reference: 65906; Sigma-Aldrich) for an overnight period at 4°C was followed by three washes in PBS. Sections were mounted with prolong gold antifade reagent (Thermo Fisher Scientific).

For electron microscopy, tadpoles were fixed during 30–40 d at 4°C in 1.5% glutaraldehyde solution prepared in phosphate buffer, adjusted to a final osmolality of 300 mOsm/kg, pH 7.8. Tadpoles were post fixed in 1% osmium tetroxide/1.5% potassium ferricyanide in 0.1 M of sodium cacodylate buffer, dehydrated in alcohol series, and embedded in EPON resin. 1-µm-thick sections were stained with toluidine blue for the identification of the glomerular region. Ultrathin sections (60 nm) were contrasted with uranyl acetate and lead citrate and viewed under a JEOL 1010 electron microscope. Random sections were used for analysis. The area covered by endolysosomal organelles was divided by the glomerular area in images taken at 60,000 or 80,000 magnification. Four experimental groups were evaluated: control (35 images from six olfactory bulbs), tadpoles injected with peptide p2.1 (25 images from four olfactory bulbs), tadpoles injected with peptide p4.2 (32 images from six olfactory bulbs), and tadpoles injected with SPARC (43 images from two olfactory bulbs).

### STED microscopy

Single-plane STED images were acquired on a Leica TCS SP8 STED 3× (Leica Microsystems) on a DMI8 stand using a 100×/1.4NA HCS2 PL APO objective. A pulsed supercontinuum light source set at 644 nm was used for excitation and a pulsed depletion laser at 775 nm was added with no pulse delay at 20% intensity. Detection was performed with a hybrid detector (HyD) between 652 and 750 nm and gating between 0.3 and 6 ns. Scanner speed was set at 600 Hz and images were taken with 8× frame accumulation and 2× frame average. Pixel size was set according to the depletion power to 23 nm.

### Statistical analysis

Statistical analysis was carried out using parametric or non-parametric tests after evaluating the experimental groups with the Kolmogorov–Smirnov test. Differences among groups that contained non-normally distributed data were established using the Kruskal–Wallis test followed by Dunn's multiple comparisons test. The differences among groups that passed the normality test were established by one-way ANOVA followed by Dunnett's multiple comparisons

test. Differences between two groups were established by the unpaired $t$ test. $P$-values < 0.05 were considered as significant.

## Data Availability

The data that support the findings of this study are included in the article and supplementary information and are available from the corresponding author upon reasonable request.

## Supplementary Information

## Acknowledgements

This work was sponsored by the Ministry of Science, Innovation and Universities, grant RTI2018-096948-B-100 (A Llobet), co-funded by the European Regional Development Fund. We thank the institutional support from the María de Maeztu Unit of Excellence, Institute of Neurosciences, University of Barcelona, MDM-2017-0729 (Ministry of Science, Innovation and Universities), and the CERCA Program of Generalitat de Catalunya. A Llobet is a Serra Húnter fellow.

### Author Contributions

B Terni: conceptualization, data curation, formal analysis, investigation, methodology, and writing—original draft, review, and editing.
A Llobet: conceptualization, data curation, formal analysis, supervision, funding acquisition, investigation, methodology, and writing—original draft, review, and editing.

### Conflict of Interest Statement

The authors declare that they have no conflict of interest.

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
