## [Reviewer comments · Life Science Alliance]

Life Science Alliance

Axon terminals control endolysosome diffusion to support synaptic remodeling

Beatrice Terni and Artur Llobet

DOI: <https://doi.org/10.26508/lsa.202101105>

Corresponding author(s): Artur Llobet, Institute of Neurosciences, University of Barcelona

Review Timeline:

Submission Date:	2021-04-22
Editorial Decision:	2021-05-20
Revision Received:	2021-06-07
Editorial Decision:	2021-06-09
Revision Received:	2021-06-14
Accepted:	2021-06-15

Transaction Report:

Please note that the manuscript was previously reviewed at another journal and the reports were taken into account in the decision-making process at Life Science Alliance. Since the original reviews are not subject to Life Science Alliance's transparent review process policy, the reports and author response cannot be published.

May 20, 2021

Re: Life Science Alliance manuscript #LSA-2021-01105-T

Dr. Artur Llobet
Institute of Neurosciences, University of Barcelona
Pavello de Govern-Lab.4112 Feixa Llarga s/n
L'Hospitalet de Llobregat, Catalonia 08907
Spain

Dear Dr. Llobet,

Thank you for submitting your manuscript entitled "Axon terminals control endolysosome diffusion to support synaptic remodeling" to Life Science Alliance.

Your manuscript has been evaluated by LSA editor and our academic board expert. We have gone through the revised manuscript, both rounds of reviews from the previous journal and the author's pbp response to the reviewers' comments.

In our opinion, the findings in the manuscript are solid, and the authors' have addressed all the points raised by the referees appropriately. We would like to invite you to submit a revised version of this manuscript to be published in LSA. The one remaining concern pointed by Rev 3, asking the authors to confirm that latrunculin A injection impairs the olfactory-driven behavior like SPARC, should be included in the revision - we might need to reach out to an expert to review this new data, in which case, we will walk them through the transfer process.

Thank you for this interesting contribution to Life Science Alliance. We are looking forward to

receiving your revised manuscript.

Sincerely,

Shachi Bhatt, Ph.D.
Executive Editor
Life Science Alliance
<http://www.lsjournal.org>
Tweet @SciBhatt @LSAJournal

- A letter addressing the reviewers' comments point by point.
- An editable version of the final text (.DOC or .DOCX) is needed for copyediting (no PDFs).
- High-resolution figure, supplementary figure and video files uploaded as individual files: See our detailed guidelines for preparing your production-ready images, <https://www.life-science-alliance.org/authors>
- Summary blurb (enter in submission system): A short text summarizing in a single sentence the study (max. 200 characters including spaces). This text is used in conjunction with the titles of papers, hence should be informative and complementary to the title and running title. It should describe the context and significance of the findings for a general readership; it should be written in the present tense and refer to the work in the third person. Author names should not be mentioned.

B. MANUSCRIPT ORGANIZATION AND FORMATTING:

June 9, 2021

RE: Life Science Alliance Manuscript #LSA-2021-01105-TR

Dr. Artur Llobet
Institute of Neurosciences, University of Barcelona
Pavelló de Govern-Lab.4112 Feixa Llarga s/n
L'Hospitalet de Llobregat, Catalonia 08907
Spain

Dear Dr. Llobet,

Thank you for submitting your revised manuscript entitled "Axon terminals control endolysosome diffusion to support synaptic remodeling". We would be happy to publish your paper in Life Science Alliance pending final revisions necessary to meet our formatting guidelines.

- please label the abstract
- please consult our manuscript preparation guidelines <https://www.life-science-alliance.org/manuscript-prep> and make sure your manuscript sections are in the correct order;
- please add an Author Contributions section to your main manuscript text
- please add a conflict of interest statement to your main manuscript text
- please add your movie legends to the main manuscript text after the figure legends
- please add a callout for Figure 2B to your main manuscript text

FIGURE CHECKS:

- missing scale bars for figure 2A

A. FINAL FILES:

-- High-resolution figure, supplementary figure and video files uploaded as individual files: See our detailed guidelines for preparing your production-ready images, <https://www.life-science->

alliance.org/authors

B. MANUSCRIPT ORGANIZATION AND FORMATTING:

Sincerely,

June 15, 2021

RE: Life Science Alliance Manuscript #LSA-2021-01105-TRR

Dr. Artur Llobet
Institute of Neurosciences, University of Barcelona
Pavelló de Govern-Lab.4112 Feixa Llarga s/n
L'Hospitalet de Llobregat, Catalonia 08907
Spain

Dear Dr. Llobet,

Thank you for submitting your Research Article entitled "Axon terminals control endolysosome diffusion to support synaptic remodeling". It is a pleasure to let you know that your manuscript is now accepted for publication in Life Science Alliance. Congratulations on this interesting work.

DISTRIBUTION OF MATERIALS:

Again, congratulations on a very nice paper. I hope you found the review process to be constructive and are pleased with how the manuscript was handled editorially. We look forward to future exciting submissions from your lab.

Sincerely,
